# Positive Regulatory Roles of *Manihot esculenta* HAK5 under K^+^ Deficiency or High Salt Stress

**DOI:** 10.3390/plants13060849

**Published:** 2024-03-15

**Authors:** Minghua Luo, Jing Chu, Yu Wang, Jingyan Chang, Yang Zhou, Xingyu Jiang

**Affiliations:** 1Key Laboratory for Quality Regulation of Tropical Horticultural Crops of Hainan Province, School of Life and Health Sciences, Hainan University, Haikou 570228, China; luominghua98@163.com (M.L.); chujing2024@163.com (J.C.); wangyuhz0718@163.com (Y.W.); 2National Center for Technology Innovation of Saline-Alkali Tolerant Rice, College of Coastal Agricultural Sciences, Guangdong Ocean University, Zhanjiang 524088, China; changjy1231@163.com

**Keywords:** cassava, high-affinity potassium transporter, potassium starvation, salt tolerance

## Abstract

HAK/KUP/KT family members have been identified as playing key roles in K^+^ uptake and salt tolerance in numerous higher plants. However, their functions in cassava (*Manihot esculenta* Cantz) remain unknown. In this study, a gene encoding for a high-affinity potassium transporter (*MeHAK5*) was isolated from cassava and its function was investigated. Subcellular localization analysis showed that *Me*HAK5 is a plasma membrane-localized transporter. RT-PCR and RT-qPCR indicated that *MeHAK5* is predominantly expressed in cassava roots, where it is upregulated by low potassium or high salt; in particular, its highest expression levels separately increased by 2.2 and 2.9 times after 50 µM KCl and 150 mM NaCl treatments. When heterologously expressed in yeast, *Me*HAK5 mediated K^+^ uptake within the cells of the yeast strain CY162 and rescued the salt-sensitive phenotype of AXT3K yeast. *MeHAK5* overexpression in transgenic *Arabidopsis* plants exhibited improved growth and increased shoot K^+^ content under low potassium conditions. Under salt stress, *MeHAK5* transgenic *Arabidopsis* plants accumulated more K^+^ in the shoots and roots and had reduced Na^+^ content in the shoots. As a result, *MeHAK5* transgenic *Arabidopsis* demonstrated a more salt-tolerant phenotype. These results suggest that *Me*HAK5 functions as a high-affinity K^+^ transporter under K^+^ starvation conditions, improving K^+^/Na^+^ homeostasis and thereby functioning as a positive regulator of salt stress tolerance in transgenic *Arabidopsis*. Therefore, *Me*HAK5 may be a suitable candidate gene for improving K^+^ utilization efficiency and salt tolerance.

## 1. Introduction

Saline soils are crucial for one-third of global food production, encompassing 6% of the world’s total land area, with an additional 20% comprised of irrigated soils affected by salt. This global issue of soil salinity significantly impacts agricultural sustainability and profitability [1,2,3]. Sodium ions (Na^+^) predominate in most saline soil regions, posing challenges as plants absorb Na^+^ from the soil, leading to its accumulation and ionic imbalances within cells. Such imbalance inhibits the uptake of essential nutrients, disrupting key metabolic processes [4].

Potassium (K^+^) is an essential macronutrient for plant growth and development, playing crucial roles in many physiological and biochemical processes, including osmoregulation, enzyme activation, charge neutralization, and membrane potential maintenance in plant cells [5,6]. K^+^ is additionally involved in regulating photosynthesis, starch synthesis, and subsequent carbohydrate transport and metabolism [7,8]. Sodium shares similar physical and chemical traits with potassium (K^+^); thus, salinity stress significantly affects K^+^ homeostasis by inhibiting K^+^ uptake into plant cells, leading to K^+^ nutrient imbalance in salt-stressed plants, which adversely affects plant growth and development [9]. Therefore, enhanced K^+^ uptake in environments with high salinity can effectively reduce or eliminate the adverse effects of salt stress [10,11,12,13].

Soil K^+^ acquisition by the plant root system and subsequent K^+^ transport across plant organs and tissues is mainly regulated by K^+^ channels and K^+^ transporters [14]. The HAK/KUP/KT (high-affinity K^+^/K^+^ uptake/K^+^ transporter) family is among the most essential K^+^ acquisition components in plants [15]. The activation and maintenance of high-affinity K^+^ uptake capacity under salt stress, mediated by the HAK/KUP/KT family, is particularly important for the maintenance of adequate intracellular K^+^ concentration or optimal K^+^/Na^+^ ratio, in turn conducive to plant survival [16]. The HAK/KUP/KT family is a widely distributed family present in bacteria, fungi, and plants. It is the largest family of K^+^ transporters in plants, and is commonly associated with the transmembrane transport of K^+^ [17]. The barley *Hv*HAK1 and the *Arabidopsis At*KUP1 were the first isolated plant HAK/KUP/KT family members based on their homology to bacterial KUP and fungal HAK transporters [18,19]. Many HAK/KUP/KT family members have been identified through genome sequencing projects in different species, and multiple members have been identified in their genomes. For example, 13 and 27 members have been identified in the model plants *Arabidopsis thaliana* and *Oryza sativa*, respectively [20,21], 27 members in *Zea mays* [22], 56 members in *Triticum aestivum* [23], 29 members in *Glycine max* [24], and 21 members in *Manihot esculenta* [25]. While plant HAK/KUP/KT proteins are predominantly localized at the plasma membrane, they present in different subcellular compartments as well, such as the tonoplast and the endomembrane system [26,27,28,29,30]. Plant HAK/KUP/KTs exhibit considerable variability in gene expression patterns, and are expressed in various tissues such as the roots, leaves, fruits, and seeds [29,31,32,33]. Plant HAK/KUP/KT family members not only function as K^+^ uptake transporters, including high- and low-affinity transporters [34,35], they play roles in K^+^ translocation, salinity, drought tolerance, and plant development [17,36]. Plant HAK/KUP/KT proteins typically contain 10–15 transmembrane domains and a K^+^ transport domain (K^+^_trans), with three cytoplasmic domains located at the N-terminus, the C-terminus, and the loop region between the second and third transmembrane domains (loop II–III), while their C-terminus is longer compared to the N-terminus [22,37]. A conserved GVVYGD motif is present in the first transmembrane domain (TM1) of the HAK/KUP/KT protein, similar to the conserved GYGD motif of the plant shaker-like K^+^-selective transporters. Based on structural modeling and mutagenesis, residues D72, D201, and E312 at TM1, TM3, and TM6 have been shown to be critical for the proteins’ transporter function [38]. Furthermore, HAK/KUP/KT family members are classified into subgroups I–IV based on their phylogenetic relationships [35].

Many plant HAK/KUP/KT family members have been cloned and functionally analyzed in different species. The barley *HvHAK1* was the first to be isolated and characterized, and has been shown to be transcriptionally induced by K^+^ deficiency and transiently induced under high salt stress. *Hv*HAK1 mediates high-affinity K^+^ uptake and low-affinity Na^+^ uptake when heterologously expressed in yeast. When *HvHAK1* was overexpressed in *Arabidopsis thaliana*, it was found to mediate K^+^ uptake under K^+^-deficiency conditions. *HvHAK1* overexpression has been demonstrated to enhance drought tolerance [18,39,40]. Yeast heterologously expressing the *Hordeum brevisubulatum HbHAK1*, a homolog of *HvHAK1*, exhibited high affinities for K^+^ uptake under very low K^+^ concentration, and its expression in an *Arabidopsis athak5* mutant enhanced salt tolerance under high salinity [41]. *At*HAK5 is the best-characterized *Arabidopsis* HAK/KUP/KT family member. *AtHAK5* transcription is induced by K^+^ starvation, and although salt stress dramatically inhibits *AtHAK5* expression, *At*HAK5 remains the major potassium acquisition component in *Arabidopsis* under low K^+^ conditions in the presence of NaCl, being the only transporter that mediates K^+^ uptake at external K^+^ concentrations below 10 µM. Seedling establishment and postgermination growth of the *Arabidopsis athak5* mutant are significantly suppressed under low K^+^ concentrations in the growth medium [42,43,44]. In *Thellungiella halophila* (a close relative of *Arabidopsis*), *ThHAK5*, when heterologously expressed in yeast, was shown to be involved in high-affinity K^+^ uptake [45]. In rice, OsHAK5 is a high-affinity K^+^ transporter and mediates K^+^ transport from the root to the shoot, and positively regulates rice salt tolerance [33]. Surprisingly, OsHAK1, OsHAK16, and OsHAK21, also subgroup I members, have similar functions to those of OsHAK5 [13,46,47]. Furthermore, SiHAK1 from *Setaria italica* was shown to be involved in high-affinity K^+^ uptake when heterologously expressed in yeast and when overexpressed in *Arabidopsis athak5* mutants [48]. Recently, CeqHAK6 from *Casuarina equisetifolia* was shown to be involved in high-affinity K^+^ uptake, and *CeqHAK6* overexpressing *Arabidopsis* plants exhibited improved salt tolerance [49]. This suggests that the plant HAK/KUP/KT family subgroup I members directly regulate K^+^ and Na^+^ homeostasis by mediating K^+^ uptake and transport, which has important implications for plant salt tolerance.

Cassava (*Manihot esculenta* Crantz) is one of the world’s most important tuber-bearing crops. In addition to being a staple crop for over a billion people, it is used as livestock feed and its biomass is used for renewable energy production. It is reported to be one of the major income sources for farmers in most tropical countries [50,51,52]. Cassava’s inherent tolerance and adaptability under poor soil and environmental conditions have made it popular among resource-limited subsistence farmers, where it can be cultivated to generate income and ensure food security in developing countries. However, cassava requires a high soil potassium content for sufficient productivity, and potassium deficiency has been shown to significantly limit cassava yield and quality [53]. Although 21 HAK/KUP/KT family members have been identified in cassava, their physiological functions have not yet been investigated [25]. Here, we have isolated and characterized a *HAK5*-like member, *MeHAK5*, from cassava. Its tissue expression pattern and transcript levels were assessed in response to salt stress and low K^+^ and it was functionally characterized in yeast and *Arabidopsis*. The results indicated that the MeHAK5 transporter performs high-affinity activity K^+^ uptake, regulates K^+^/Na^+^ homeostasis, and enhances salt tolerance under salinity stress conditions.

## 2. Materials and Methods

### 2.1. Plant Materials and Growth Conditions

Cassava (*Manihot esculenta* Crantz Luoyong 9) stem segments of about 1.5 cm containing axillary buds cut from sixty-day-old plants were placed in 1/2 MS solid medium (pH = 5.8) and grown in a plant growth chamber for 50 days under the following conditions: long day photoperiod (16 h light/8 h dark), temperature of 28 °C, 70% relative humidity. Healthy and uniformly sized cassava seedlings were selected and acclimated in an Afdaling nutrient solution [54]. After one week of acclimation, they were used for the subsequent experiments.

*Arabidopsis thaliana* ecotype Columbia (Col-0) plants were used as the WT; its seeds, which are kept in our laboratory, were surface sterilized with 10% sodium hypochlorite, placed for 3 days at 4 °C in the dark for vernalization, then sown on 1/2 MS solid medium and grown vertically in an illuminated plant growth chamber under the following conditions: 22 °C, 16 h light/8 h dark, and 70% relative humidity. Five-day-old seedlings were transplanted to pots (diameter: 7 cm, height: 9 cm) filled with vermiculite and peat (1:1, *v*/*v*) with a density of five plants per pot and grown under the conditions mentioned above.

### 2.2. Isolation and Bioinformatic Analysis of MeHAK5

Total RNA extraction from cassava roots was performed using a RNAprep Pure Plant Kit (TIANGEN, Beijing, China) according to the manufacturer’s instructions. RNA quality and integrity were evaluated using an Agilent 2100 Bioanalyzer (Agilent Technologies, Inc., Santa Clara, CA, USA) and agarose gel electrophoresis. First-strand cDNA was synthesized using a PrimeScript II 1st Strand cDNA Synthesis Kit (TaKaRa, Maebashi, Japan) following the included protocol instructions. The cDNA was used as a template for amplification of the target gene with specific primers (Appendix A). The PCR products were cloned into T vectors using the pEASY^®^ Blunt Cloning Kit (Trans Gen, Beijing, China), and sequencing was performed for verification.

*Arabidopsis* HAK5 (AtHAK5, NP_567404.1) functions as a high-affinity K^+^ transporter, and is the best-characterized *Arabidopsis* HAK/KUP/KT family member [42,43,44]. In light of this, AtHAK5 was used to capture HAK5-type transporters from the NCBI website via BLAST searching (http://www.ncbi.nlm.nih.gov/gorf/gorf.html, accessed on 16 October 2022). A HAK5-type gene (*MeHAK5*) was found on the website. Multiple sequence alignment of the MeHAK5 protein was performed with DNAMAN 6.0 software. Predictions of the transmembrane structural domains of the MeHAK5 protein were obtained using the online software DeepTMHMM (https://dtu.biolib.com/app/DeepTMHMM/run, accessed on 18 November 2023). The phylogenetic tree was constructed using the neighbor-joining method in MEGA 11.0 software, and bootstrap values were set to 1000 replicates.

### 2.3. MeHAK5 Expression Analyses

Total RNA from different cassava tissues (roots, stems, and leaves) was extracted, the synthesized cDNA from the extracted RNA was used as the template for PCR amplification with the gene-specific primers *MeHAK5*-semiF and *MeHAK5*-semiR (Appendix A), and *MeACTIN* was used as the internal reference gene to assess the *MeHAK5* tissue expression patterns. The PCR reaction conditions were as follows: initial denaturation at 95 °C for 3 min, followed by 28 cycles of 94 °C for 15 s, 56 °C for 15 s, and 72 °C for 10 s, followed by extension at 72 °C for 5 min. For further *MeHAK5* gene expression quantification, cassava plants were treated with 50 µM K^+^ or 150 mM NaCl. Their roots were harvested at 0, 3, 6, 12, 24, or 48 h after stressed treatments, then immediately placed in liquid nitrogen and stored at −80 °C. RNA was extracted from the frozen samples. Real-time quantitative PCR (qRT-PCR) was performed using RealUniversal Color PreMix SYBR Green (TIANGEN, Beijing, China) with *MeHAK5*-specific primers (Appendix A) and *MeACTIN* as the internal reference gene. RT-qPCR was performed in a Qiagen Rotor Gene Q real-time PCR instrument (Qiagen, Hiden, Germany) under the following cycling conditions: predenaturing at 95 °C for 15 min, followed by 40 cycles at 95 °C for 10 s and 60 °C for 30 s. The comparative CT method (2^−ΔΔCT^) method was used to calculate the relative expression levels of the *MeHAK5* gene.

### 2.4. Tobacco Transformation and Subcellular Localization Analysis

To determine the subcellular location of MeHAK5, its complete coding sequence (without the stop codon) was amplified using the primers *MeHAK5*-*Sma* I-R and *MeHAK5*-*Bam*H I-F (Appendix A). The PCR products and the modified plant vector pCAMBIA1300-GFP were double-digested with *Bam*H I and *Sma* I, respectively. To generate fused *MeHAK5-GFP* gene, the target gene fragments and large vector fragments were recovered and ligated to construct the recombinant vector pCAMBIA1300-MeHAK5-GFP, as shown in Appendix A. The recombinant and empty pCAMBIA1300-GFP plasmids were transformed into *Agrobacterium tumefaciens* strain GV3101, which was then injected into tobacco (*Nicotiana benthamiana*) leaves. GFP fluorescence was observed 4 days after inoculation using a fluorescence confocal microscope (FV3000; Olympus Corporation, Tokyo, Japan) as previously reported [55].

### 2.5. MeHAK5 Functional Complementation Assays in Yeast

The full-length *Me*HAK5 coding sequence was amplified with gene-specific primers (Appendix A), digested with the *Bam*H I/*Sma* I restriction endonucleases, and inserted into the yeast expression vector p416 to generate the vector p416-MeHAK5, as shown in Appendix A. The recombinant vectors and the empty p416 vector were transferred into the yeast strains CY162 and AXT3K by the LiAc/PEG transformation method. For the yeast functional compensation assay, mobilized yeast cells harboring p416 and MeHAK5-p416 were transferred to a YPD liquid medium and incubated to saturation at 28 °C and 200 rpm. The cultural solution was diluted ten times, then 5 μL aliquots of each serial dilution were spotted onto AP plates supplemented with the indicated concentrations of KCl or NaCl and allowed to grow for 3–5 days at 28 °C.

### 2.6. Functional Analysis of MeHAK5 in Arabidopsis thaliana

The *MeHAK5* open reading frame (ORF) was amplified with gene-specific primers (Appendix A), digested with *Spe* I/*Bam*H I restriction endonuclease, and cloned into the pCAMBIA1300 plant expression vector to construct the recombinant vector pCAMBIA1300-MeHAK5 (Appendix A). The recombinant vector was transformed into *Agrobacterium tumefaciens* strain GV3101, which was then used for *Arabidopsis thaliana* transformation as described previously [56]. Positive transformants were obtained by screening with 1/2 MS containing 50 mg/L hygromicin B and further verified by PCR amplification. T3 homozygous transgenic lines were generated and used in the subsequent experiments.

WT and transgenic *Arabidopsis* seeds were sown in 1/2 MS medium. After 4 days of growth, uniformly sized seedlings were transferred to a fresh 1/2 MS solid medium containing 10 µM KCl or 75 mM NaCl and allowed to grow at 22 °C with a 16 h light and 8 h dark photoperiod for 14 days. The plants were then photographed and their different growth and physiological parameters were determined.

To further verify the performance of *MeHAK5*-overexpressing transgenic *Arabidopsis* in the soil in response to salt stress, WT and transgenic plants seeds were sown on 1/2 MS medium. Five-day-old seedlings of equal size were transplanted to the soil and grown in a greenhouse at 22 °C with a 12 h light and 12 h dark photoperiod. After 8 days, the seedlings were treated with salt by watering with 2 L of a 300 mM NaCl solution, while control seedlings were watered with the same volume without NaCl. The corresponding parameters and growth phenotypes were determined and photographed after 3 and 5 weeks of salinity treatment, respectively.

### 2.7. Ion Content Determination

Thirteen-day-old seedlings grown in soils were treated with 300 mM NaCl and allowed to grow for 3 weeks. After that, the shoots and roots were separately harvested and washed with distilled water, then dried to a constant weight in an oven at 80 °C. The dried samples were ground and digested with 100 mM HNO_3_ (1:60, *w*/*v*) for 12 h and then diluted with deionized water. The K^+^ and Na^+^ concentrations in the extract solution were determined by inductively coupled plasma optical emission spectroscopy (ICP-OES; PerkinElmer) as previously reported [57].

### 2.8. Determination of Chlorophyll Content in Leaves

The leaves were ground with liquid nitrogen and mixed with 80% acetone. The mixture was left at room temperature overnight in the dark with shaking, followed by centrifugation at 12,000 rpm for 10 min. Absorbance values in the supernatant were determined at 663 nm and 645 nm using a spectrophotometer (L6, Yoke Instrument, Shanghai, China). Chlorophyll concentration was calculated as described previously [58].

### 2.9. Determination of Proline Content in Leaves

The proline content in leaves was determined using proline assay kits (Nanjing Jiancheng Bioengineering Institute, Nanjing, China) according to the manufacturer’s instructions. Fresh leaves were homogenized in Reagent I (1:9, *w*/*v*) on ice. The homogenate underwent centrifugation at 3500 r/min for 10 min. Subsequently, the supernatant was mixed with Reagent II and Reagent III (1:2:2, *v*/*v*/*v*), then the reaction proceeded at 100 °C in a water bath for 30 min. After cooling, the proline content in the extract solution was determined using an enzyme-labeling instrument (Infinite M200 PRO, TECAN Corporation, Männedorf, Switzerland) as described in the manufacturer’s instructions.

### 2.10. Determination of Malondialdehyde (MDA) Content in Leaves

The MDA content in leaves was determined using malondialdehyde assay kits (Nanjing Jiancheng Bioengineering Institute) according to the manufacturer’s instructions. Fresh leaves were homogenized in extract solution (1:9, *w*/*v*) on ice, followed by centrifugation at 4000 rpm for 10 min. The resulting supernatant was combined with working solution (1:20, *v*/*v*) and then incubated at 100 °C in a water bath for 20 min. After cooling, the MDA content in the reaction solution was quantified using an enzyme-labeling instrument as described in the manufacturer’s instructions.

### 2.11. Statistical Analysis

All experiments were performed in three biological replicates unless otherwise stated. All data were assayed using Tukey’s test (*p* < 0.05) and one-way ANOVA with SPSS 20 software. Data are expressed as mean ± standard deviation (SD), and *p* < 0.05 was regarded as statistically significant.

## 3. Results

### 3.1. Isolation and Characterization of MeHAK5 from Cassava

The 2427 bp full-length *MeHAK5* CDS was isolated from cassava; the deduced *Me*HAK5 protein contained 808 amino acids, with a predicted protein molecular weight of 89.88 KDa and a predicted theoretical isoelectric point of 8.89. The *Me*HAK5 protein was predicted to contain twelve transmembrane domains (Appendix A). The deduced MeHAK5 protein sequence was aligned with other HAK5 proteins in plants, and shared an 89, 79, 62, and 59% homology identity with its *Hevea brasiliensis*, *Ricinus communis*, *Arabidopsis thaliana*, and *Oryza sativa* homologues, respectively (Figure 1A). The deduced *Me*HAK5 protein contained a K^+^ transport domain (K_trans), unique to the HAK/KUP/KT family, along with a conserved GVVYGD motif in the first transmembrane domain (TM1) shared by the HAK/KUP/KT family, which may be under evolutionary selection pressure, resulting in the unique catalytic properties of each protein. Furthermore, the *Me*HAK5 protein contained the conserved residues D72, D201, and E312 in TM1, TM3, and TM6, respectively, which are thought to be involved in K^+^ binding, while G67, Y70, and G71 are hypothesized to be responsible for K^+^ selection. Therefore, the six amino acid residues are essential for K^+^ transport by members of the plant HAK/KUP/KT family. In addition, F130 is predicted to affect the affinity of HAK5 for K^+^ [38,59].

Phylogenetic tree analyses showed that HAK5 proteins can be divided into two clusters: ZmHAK5, SbHAK5, OsHAK5, TaHAK5, and HvHAK5 from monocotyledons cluster together, while MeHAK5, HbHAK5, RcHAK5, PaHAK5, TwHAK5, VvHAK5, JrHAK5, PvHAK5, GmHAK5, MtHAK5, and AtHAK5 from dicotyledons form another cluster. As seen from the phylogenetic tree, MeHAK5 is genetically more closely related to HAK5s from dicotyledonous plants than to HAK5s of monocotyledonous plants. This similarity is particularly evident in *Hevea brasiliensis* and *Ricinus communis*, which along with cassava are members of the *Euphorbiaceae* family (Figure 1B; Appendix A). These results indicate that MeHAK5 may have K^+^ uptake or transport activity.

### 3.2. Expression Analysis of the Cassava MeHAK5 Gene

Based on RT-PCR, *MeHAK5* was significantly expressed in the roots, while it was not detected in the stems or leaves (Figure 2A). *MeHAK5* expression in response to low potassium or high salt stress was further investigated by RT-qPCR. *MeHAK5* expression in the roots treated with 50 µM KCl initially increased gradually and reached a peak at 12 h, 2.2-fold higher compared to the 0 h time point. It then gradually decreased, and at 48 h returned to the expression level measured at 0 h (Figure 2B). However, under 150 mM NaCl condition *MeHAK5* was rapidly upregulated in the roots and reached a peak at 3 h, a 2.9-fold increase compared to 0 h, then remained relatively stable until 24 h and at 48 h returned to a level similar to that at 0 h (Figure 2C). The above results indicate that *MeHAK5* is mainly expressed in the roots and that its transcript levels are upregulated by low potassium stress or high salt stress.

### 3.3. Subcellular Localization of the MeHAK5 Protein

The *Me*HAK5 protein was predicted to have twelve transmembrane domains, suggesting that it is a membrane-localized protein. Its in vivo subcellular localization was further investigated. The recombinant vector pCAMBIA1300-MeHAK5-GFP was constructed and introduced into *A. tumefaciens*, which was injected into the epidermal cells of *N. benthamiana* leaves. Transient expression was observed 4 days after inoculation, using pCAMBIA1300-GFP as a control. The GFP fluorescence signal in the control was observed in the cytosol. In contrast, green fluorescence from the MeHAK5-GFP fusion protein was detected in the plasma membrane (Figure 3). The above results indicate that the *Me*HAK5 protein is plasma membrane-localized.

### 3.4. MeHAK5 Functional Assessment in the Yeast Reconstitution System

The K^+^ uptake properties of *Me*HAK5 were investigated by heterologous expression in the yeast mutant CY162, which lacks potassium uptake proteins. Yeast expressing *MeHAK5* grew, as did transgenic yeast cells carrying the empty vector at 10 mM K^+^. However, under low potassium conditions (≤1 mM) yeast cells carrying the empty vector failed to grow, while *MeHAK5* expression rescued the inhibited growth of the yeast mutant strain. Yeast expressing *MeHAK5* grew well at 0.1 mM K^+^ (Figure 4A) and had a similar growth phenotype as at 10 mM K^+^. These results suggest that *Me*HAK5 functions as a strong K^+^ transporter under low potassium conditions.

AXT3K is a salt-sensitive mutant yeast strain that cannot grow above 30 mM NaCl concentrations [60]. Therefore, this mutant can be implemented for heterologous verification of the functions and involvement of cation transporters in salt tolerance. Recombinant p416-MeHAK5 was introduced into AXT3K and plated on AP solid medium containing different NaCl concentrations; transformants carrying the empty vector were used as control. Both *Me*HAK5 and p416 transformants were able to grow well under control conditions; however, significant differences in the growth of yeast carrying *Me*HAK5 and p416 were observed under salt stress. At a 50 mM NaCl concentration, the p416 transformants did not grow while the growth of the *MeHAK5*-expressing yeast was not obviously inhibited (Figure 4B). These results indicate that *Me*HAK5 was able to rescue the salt-sensitive phenotype of the AXT3K mutant, suggesting that it positively regulates salt tolerance under NaCl stress.

### 3.5. MeHAK5 Improves the Growth of Transgenic Arabidopsis thaliana under Potassium Starvation

To test whether MeHAK5 has a high affinity for K^+^ uptake activity under low potassium conditions, the full length of *MeHAK5* was cloned into the plasmid pCAMBIA1300, then the resulting vector pCAMBIA1300-MeHAK5 was used to transform *Arabidopsis* plants. RT-PCR showed that *MeHAK5* expression was not detected in WT but was remarkable in the two transgenic lines (Appendix A). Wild-type (WT) and *MeHAK5* transgenic *Arabidopsis* seeds were sown on 1/2 MS medium. Four-day-old uniform seedlings were transplanted to a low-potassium (LK) medium containing 10 µM K^+^ and a 1/2 MS medium (control), respectively, and were grown for 14 days. The growth phenotypes of WT and *MeHAK5* transgenic plants on 1/2 MS medium were similar. Although the growths of all tested plants were inhibited under potassium deficiency, the *MeHAK5* transgenic plants grew significantly better on the LK medium compared to the WT plants (Figure 5A). Further determination of associated parameters revealed that under LK stress the fresh weight and the K^+^ content in the shoots of *MeHAK5* transgenic plants were significantly higher than those of WT (Figure 5B), as was the length of the primary root (Figure 5C). However, under control conditions these parameters did not differ significantly between the *MeHAK5* transgenic plants and WT plants. These results suggest that the *MeHAK5* transporter has K^+^ uptake activity under K^+^-deficient conditions.

### 3.6. MeHAK5 Improves Salt Tolerance in Transgenic Arabidopsis thaliana under Salt Stress

To investigate the salt tolerance of *MeHAK5*-overexpressing transgenic *Arabidopsis* under salt stress, WT and *MeHAK5* transgenic plants were subjected to 75 mM NaCl treatment for 14 days on 1/2 MS medium. Under control conditions, no significant morphological differences were observed between the WT and *MeHAK5* transgenic plants. However, the growth of *MeHAK5* transgenic plants was significantly better than that of WT plants under 75 mM NaCl (Figure 6A). The *MeHAK5* transgenic plants had a significantly higher fresh weight and root length compared to the WT plants, which was consistent with the observed phenotypes (Figure 6B,C).

### 3.7. MeHAK5 Positively Regulates the Tolerance to Salt Stress of Transgenic Arabidopsis Grown in the Soil

To further verify whether the performance of transgenic plants in the culture medium under salt stress corresponded to improved salt stress tolerance when grown in the soil, WT and *MeHAK5* transgenic plants grown in soil were subjected to salt stress. The growth phenotypes of WT and *MeHAK5* transgenic plants were not significantly different under control conditions. However, *MeHAK5* transgenic plants exhibited significantly improved growth under salt stress compared to WT plants, which senesced and died. In contrast, *MeHAK5* transgenic plants remained green and flowered (Figure 7A). Under salt stress, the *MeHAK5* transgenic plants had a significantly greater plant height and fresh weight compared to WT. In contrast, there was no significant difference in plant height or fresh weight between *MeHAK5* transgenic plants and WT plants under control conditions (Figure 7B,C).

Next, salt stress-associated parameters were further determined in the WT and *MeHAK5* transgenic plants. Under salt stress, the Na^+^ content in shoots and roots of *MeHAK5* transgenic plants was lower than in those of WT, with a significant difference observed in the shoots (Figure 8A,B). The K^+^ content in the shoots of *MeHAK5* transgenic plants was significantly higher compared to WT, while it was only slightly higher in the roots (Figure 8C,D). As a result, the *MeHAK5* transgenic plants had a significantly lower Na^+^/K^+^ ratio in the shoots and roots compared to the WT plants (Figure 8E,F). Furthermore, the chlorophyll and proline contents of *MeHAK5* transgenic plants were significantly higher than those of WT plants under salt stress, while their MDA content was significantly lower when compared to WT (Figure 8G–I). Under control conditions, no differences were observed in the above parameters between WT and *MeHAK5* transgenic plants. These results further confirm the significantly higher salt tolerance of *MeHAK5* transgenic plants compared to WT plants under soil salt stress conditions.

## 4. Discussion

HAK/KUP/KT family members exhibit a wide range of expression patterns. The barley *HvHAK1*, the first HAK/KUP/KT family member to be isolated in higher plants, is mainly expressed in roots. *HvHAK1* transcript levels have been shown to be induced by K⁺ starvation and transiently upregulated by salt stress [18,39]. *HbHAK1* from the halophyte *Hordeum brevisubulatum* has been shown to be predominantly expressed in the roots, with its transcription being induced by salt stress [41]. The *Arabidopsis AtHAK5*, similarly expressed mainly in the roots, has been shown to be induced by potassium deficiency, while its transcription level was reduced in the presence of salt stress. Similarly, while mRNA abundance of *ThHAK5* increased in roots under potassium deficiency, its expression was downregulated by NaCl treatment [42,45]. *ZmHAK1* was found to be mainly expressed in maize shoots, with its transcription upregulated by low potassium. On the contrary, its homologue *ZmHAK5* was predominantly transcripted in roots, and its expression was not affected by low potassium [61]. In rice, *OsHAK5* was expressed in whole plant and its transcription was enhanced by K⁺ deprivation [33]. Based on the results of our study, *MeHAK5* expression could be detected only in cassava roots, where it was induced by low potassium or salt stress (Figure 2), suggesting that MeHAK5 may be mainly involved in K^+^ uptake of cassava exposed to salt stress. Therefore, it can be concluded that HAK/KUP/KT family members exhibit diverse expression patterns.

Increasing evidence suggests that the subgroup I members of the HAK/KUP/KT family are involved in K^+^ uptake. *Arabidopsis* AtHAK5 has been shown to be involved in high-affinity K^+^ uptake, being the sole contributor below 10 µM K^+^ [42,43,44], while *Th*HAK5 from *Thellungiella halophila* was shown to be involved in K^+^ uptake in yeast cells under low potassium conditions [45]. *Os*HAK1 contributed 30% and 50–55% to K^+^ uptake in rice at external potassium concentrations of 1 mM and 0.05–0.1 mM, respectively [46]. *Os*HAK5, on the other hand, exhibited high-affinity K^+^ uptake capacity under low potassium conditions [33]. *Os*HAK16 and *Os*HAK21 were shown to be reminiscent of *Os*HAK5 in K^+^ uptake under low potassium conditions [13,47]. Maize *ZmHAK1* and *ZmHAK5* were heterologously expressed in yeast and were both involved in high-affinity K^+^ acquisition under low potassium conditions. Notably, *Zm*HAK1 exhibited a lower K^+^ uptake capacity compared to *Zm*HAK5. Interestingly, *Zm*HAK1 and *Zm*HAK5 play different roles in maize under K^+^-deficient conditions, being involved in the distribution and uptake of K^+^, respectively [61]. In barley, *HvHAK1* when heterologously expressed in yeast and *Arabidopsis* was shown to function in high-affinity K^+^ uptake [18,39]. *HbHAK1* expressed in the yeast mutant CY162 exhibited a strong K^+^ uptake capacity under extreme K^+^ deficiency [41]. Furthermore, both wheat *Ta*HAK1-4A and *Casuarina equisetifolia Ceq*HAK6 have recently been shown to participate in high-affinity K^+^ uptake under K^+^-deficient conditions [49,62]. In this study, *MeHAK5* was expressed in the yeast mutant CY162. Under low potassium conditions, *Me*HAK5 was able to rescue the growth of the CY162 mutant, increasing its high-affinity K^+^ uptake (Figure 4A). *MeHAK5* expression in *Arabidopsis* further confirmed its functions related to high-affinity K^+^ uptake. Under 10 µM KCl condition, *MeHAK5* transgenic plants had better growth performance compared to WT plants, with higher biomass and shoot K^+^ content and longer primary root lengths (Figure 5), supporting the conclusion that HAK5 mainly functions in the range of 10–50 µM K^+^ concentration [55]. These results are consistent with those in transgenic *Arabidopsis* plants overexpressing *Ta*HAK1-4A [62]. Thus, *Me*HAK5 functions in relation to high-affinity K^+^ uptake under low potassium conditions.

Plants usually develop potassium deficiency under high sodium concentrations in their surrounding environment due to the physicochemical similarities between sodium and potassium, leading to increased membrane injury and decreased water content in plant cells [63]. Therefore, maintaining K^+^ uptake in salt-stressed plants is crucial for their Na^+^/K^+^ homeostasis and salt tolerance [10,17]. It is evident that increasing the expression of high-affinity HAK transporters is an effective strategy for improving K^+^ acquisition and plant growth under high salinity conditions [44]. *Arabidopsis thaliana* AtHAK5 plays an important role in high-affinity K^+^ acquisition and improved plant growth under salt stress, while the high-affinity K^+^ uptake capacity of *Th*HAK5 is an important factor in the inherent salt tolerance of the extremophile *Thellungiella halophila* [44,45]. *OsHAK5* expression in tobacco BY2 cells resulted in the accumulation of excess K^+^, but not Na^+^, during salt stress, and led to increased BY2 cell salt tolerance [64]. *OsHAK5* overexpression in rice increased the total K^+^ content in the shoots by approximately 43–115% and significantly improved the growth and the K^+^/Na^+^ ratio of the shoots, while the growth and K^+^/Na^+^ ratio of the roots were similar to those of WT plants [33]. *Os*HAK1, *Os*HAK16, and *Os*HAK21 have similar functions to *Os*HAK5, playing key roles in rice salt tolerance [12,46,47], while *Si*HAK1 from *Setaria italica* has shown the ability to rescue the salt sensitivity of the *athak5* mutant by maintaining K^+^ uptake under salt stress [48]. Recently, *Hb*HAK1 from *Hordeum brevisubulatum* has been found to improve salt tolerance in the same *athak5* mutant by increasing K^+^ uptake [41]. In this study, transgenic *Arabidopsis* plants overexpressing *MeHAK5* accumulated more K^+^ and less Na^+^, especially in the shoots, resulting in a significantly lower Na^+^/K^+^ ratio in the shoots and roots of transgenic plants under salt stress compared to WT. Correspondingly, transgenic plants exhibited more robust growth and overall vigor under salt stress (Figure 6, Figure 7 and Figure 8). These results are consistent with those after overexpression of rice *OsHAK1* and *Casuarina equisetifolia CeqHAK6* in transgenic plants, which have been shown to accumulate more K^+^ and less Na^+^, resulting in significantly lower Na^+^/K^+^ ratios and greater salt tolerance [49,65]. It has been reported that the contents of chlorophyll and proline in *Acacia ampliceps* were significantly increased under salt stress [66]. Furthermore, the salt-tolerant sugarcane cultivar K88-92 accumulated more proline relative to salt-sensitive cultivars under NaCl treatment; correspondingly, its photosynthetic rate was higher than salt-sensitive cultivars [67]. *OsHAK1*-trangenic rice plants, which are more salt tolerant than wild-type plants, accumulated less MDA relative to wild type plants under salt stress [65]. Thus, the changes in the contents of chlorophyll, proline, and MDA in plants exposed to salt stress can effectively reflect the differences in salt tolerance among plants. In the present study, the transgenic plants with *MeHAK5* had significantly higher chlorophyll and proline contents compared to the wild-type plants, indicating that the transgenic plants maintained more efficient photosynthesis and better osmotic stress resistance (Figure 8G,H). On the other hand, the MDA content was significantly lower than that of the wild type, suggesting that the membrane lipids of the transgenic plants were more effectively protected from oxidative stress than those of the WT plants (Figure 8I). The above results indicate that *Me*HAK5 can improve K^+^ and Na^+^ homeostasis under salt stress, thereby enhancing salt tolerance.

## 5. Conclusions

The development of salt-tolerant crop varieties is an effective solution for dealing with soil salinization and addressing sustainable agricultural development [68]. For example, while *OsHAK1* overexpression does not affect the growth of transgenic rice under normal conditions, rice plants overexpressing *OsHAK1* show increased yields by 25% under salt stress compared to the wild type [65]. In the present study, we found that cassava *MeHAK5* is mainly expressed in the roots, where it is upregulated under low potassium or salt stress conditions. *MeHAK5* was found to be heterologously expressed in yeast and *Arabidopsis*, where it functions as a high-affinity K^+^ uptake transporter under low K^+^ conditions. *MeHAK5* overexpression in transgenic *Arabidopsis* improved K^+^ and Na^+^ homeostasis and decreased the Na^+^/K^+^ ratio under salt stress, improving the salt tolerance of transgenic plants compared to WT plants. Therefore, *MeHAK5* could be a candidate gene for breeding potassium-efficient and salt-tolerant cassava species suitable for planting in the large saline soil areas along the coastlines of Guangdong, Guangxi, and Hainan, which are the important cassava production areas in China.

## Figures and Tables

**Figure 1 plants-13-00849-f001:**
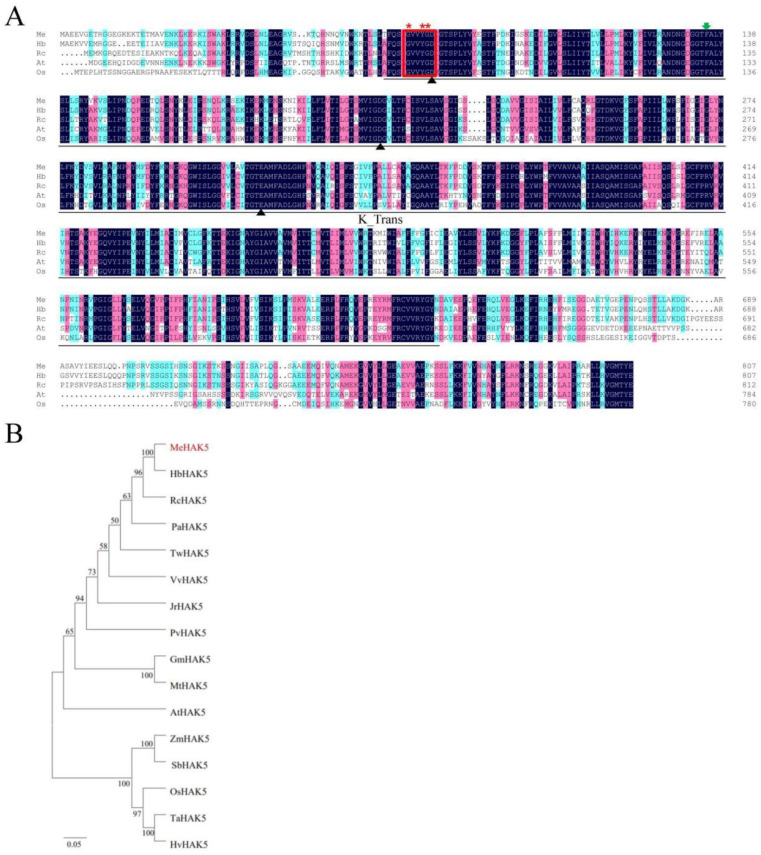
Alignment and phylogenetic analysis of the *Me*HAK5 protein. (**A**) Multiple alignment of the deduced amino acid sequences of *HAK5*-like proteins from *Manihot esculenta*, *Hevea brasiliensis*, *Ricinus communis*, *Arabidopsis thaliana,* and *Oryza sativa*. The deduced conserved motifs are indicated by red boxes. The conserved residues involved in K^+^ binding are indicated by black solid triangles, while the conserved residues involved in K^+^ selectivity is indicated by red asterisks. The K^+^ affinity is indicated by green arrows and the structural domains of K^+^ transport are indicated by lines. (**B**) Phylogenetic analysis of HAK5 proteins from different plant species (*Manihot esculenta*, *Hevea brasiliensis*, *Ricinus communis*, *Populus alba*, *Juglans regia*, *Vitis vinifera*, *Pistacia vera*, *Tripterygium wilfordii*, *Glycine max*, *Medicago truncatula*, *Arabidopsis thaliana*, *Zea mays*, *Sorghum bicolor*, *Triticum aestivum*, *Hordeum vulgare*, and *Oryza sativa*). Their accession numbers are shown in Appendix A.

**Figure 2 plants-13-00849-f002:**
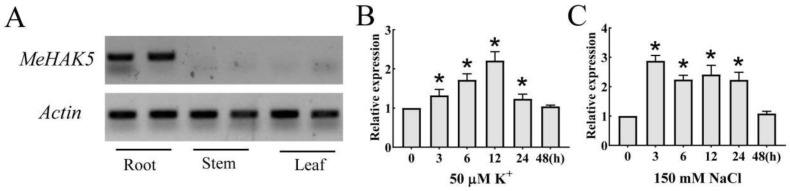
*MeHAK5* gene expression analysis in cassava. (**A**) Tissue expression patterns of the *MeHAK5* gene in cassava. (**B**) Transcript levels of the *MeHAK5* gene in cassava roots under 50 µM K^+^ concentration for 48 h. (**C**) Transcript levels of the *MeHAK5* gene in cassava roots under 150 mM NaCl concentration for 48 h. *Actin* was used as an internal control. Data are expressed as the mean ± SE of three replicates. Asterisks indicate significant differences between each time point and 0 h as indicated by Tukey’s test (*p* < 0.05).

**Figure 3 plants-13-00849-f003:**
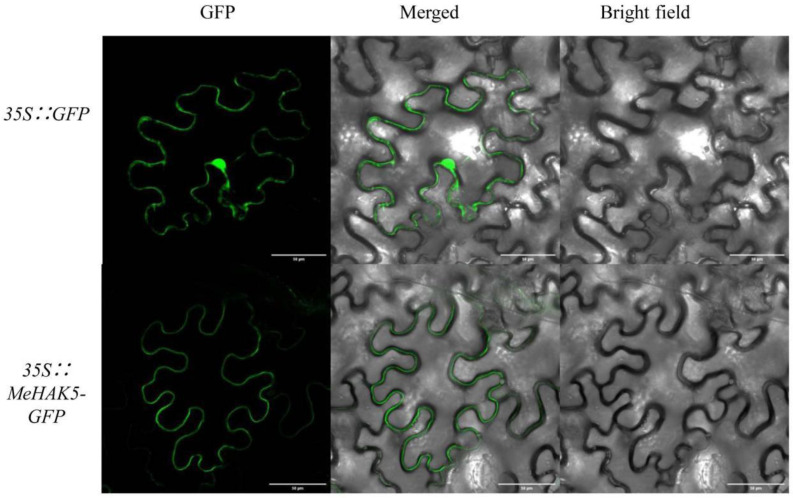
*Me*HAK5 subcellular localization in *N. benthamiana* leaf epidermal cells. The expression of *GFP* in *N. benthamiana* leaf epidermal cells was used as a reference. Scale bars on all panels in this figure correspond to 50 µm.

**Figure 4 plants-13-00849-f004:**
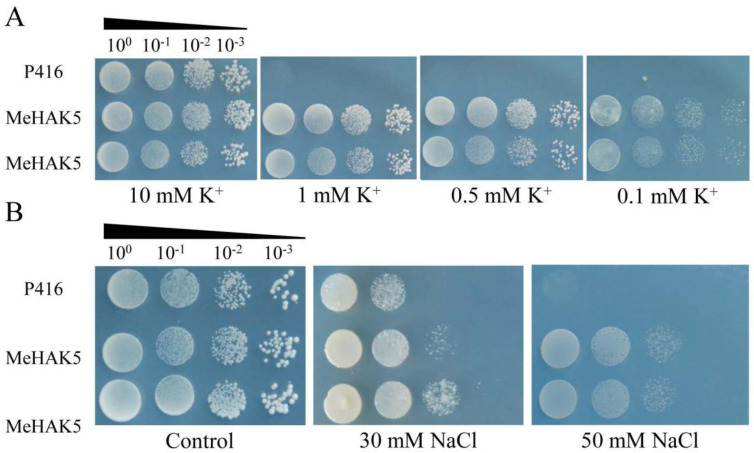
*Me*HAK5 functional complementation assays in yeast mutants. *MeHAK5* was cloned into the p416 plasmid. The resulting plasmid p416-MeHAK5 and empty vector p416 were introduced into (**A**) CY162 and (**B**) AXT3K, respectively. Transgenic yeast cells were spotted on AP plates containing different concentrations of KCl/NaCl as indicated and allowed to grow at 28 °C for 3–5 d.

**Figure 5 plants-13-00849-f005:**
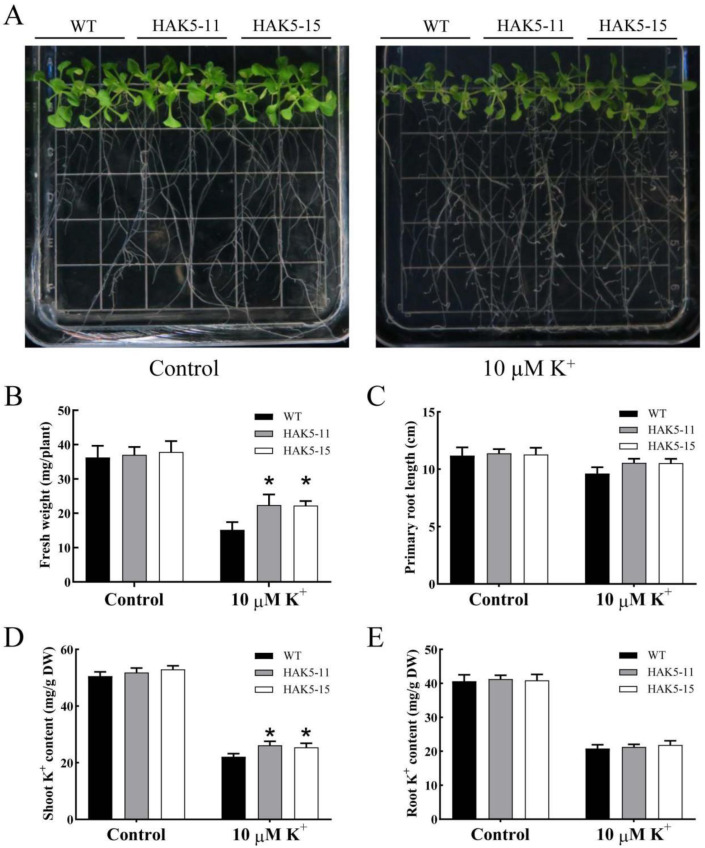
*Me*HAK5 improves *Arabidopsis* growth under K^+^ deficiency conditions. Four-day-old seedlings of WT and the two transgenic lines HAK5-11, HAK5-15 were transferred to 1/2 MS plates with or without a 10 µM K^+^ concentration. Their phenotypes (**A**) were observed on the 14th day after transfer and the (**B**) seedling fresh weight, (**C**) primary root length, and (**D**,**E**) shoot and root K^+^ levels were measured in WT and transgenic plants after K^+^ deficiency stress for 14 d. Values are expressed as means ± SD of three replicates. Asterisks indicate significant differences between WT and transgenic plants as indicated by Tukey’s test (*p* < 0.05).

**Figure 6 plants-13-00849-f006:**
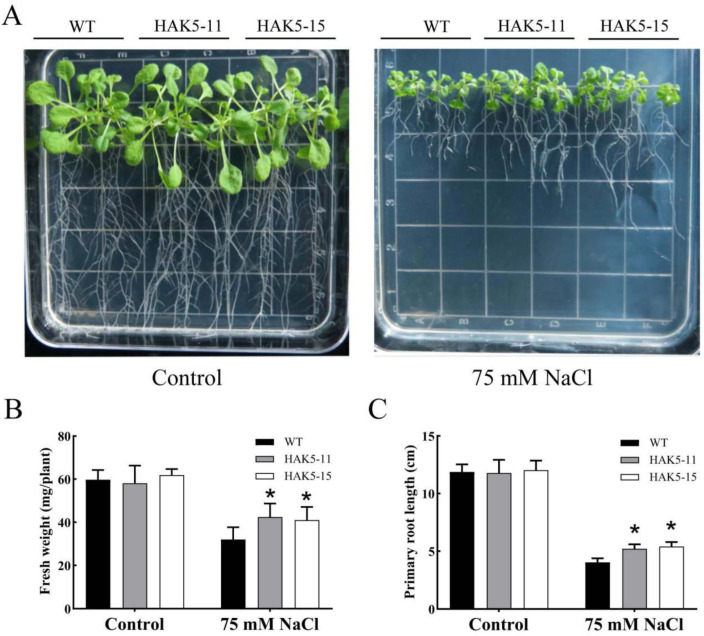
*MeHAK5* overexpression improves *Arabidopsis* growth under salt stress. Four-day-old seedlings of WT and the two transgenic lines HAK5-11, HAK5-15 were transferred to 1/2 MS plates with or without 75 mM NaCl. and their phenotypes (**A**) were observed on the 14th day after transfer. (**B**) Fresh weight of the seedlings and (**C**) length of the primary roots were measured in WT and transgenic plants grown under salt stress for 14 d. Values are expressed as means ± SD of three replicates. Asterisks indicate significant differences between WT and transgenic plants as indicated by Tukey’s test (*p* < 0.05).

**Figure 7 plants-13-00849-f007:**
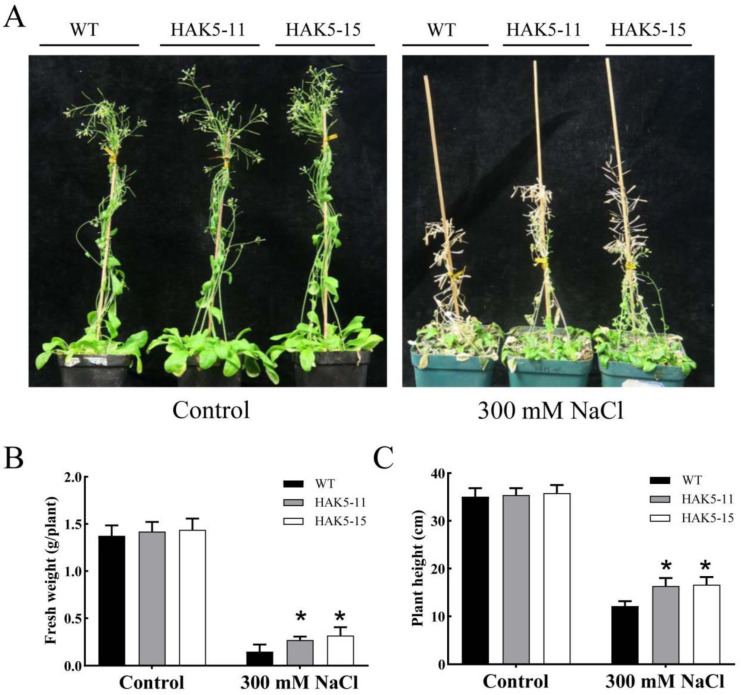
*MeHAK5* overexpression improves the growth of *Arabidopsis* plants grown in soil under salt stress. Five-day-old seedlings of WT and the two transgenic lines HAK5-11 and HAK5-15 were transferred to soil and allowed to grow for 8 days, then treated with or without 300 mM NaCl. The phenotypes (**A**) were observed on the 35th day after salt treatment and (**B**) plant fresh weight and (**C**) height were measured in WT and transgenic plants grown in the soil under salt stress for 35 d. Values are expressed as means ± SD of three replicates. Asterisks indicate significant differences between WT and transgenic plants as indicated by Tukey’s test (*p* < 0.05).

**Figure 8 plants-13-00849-f008:**
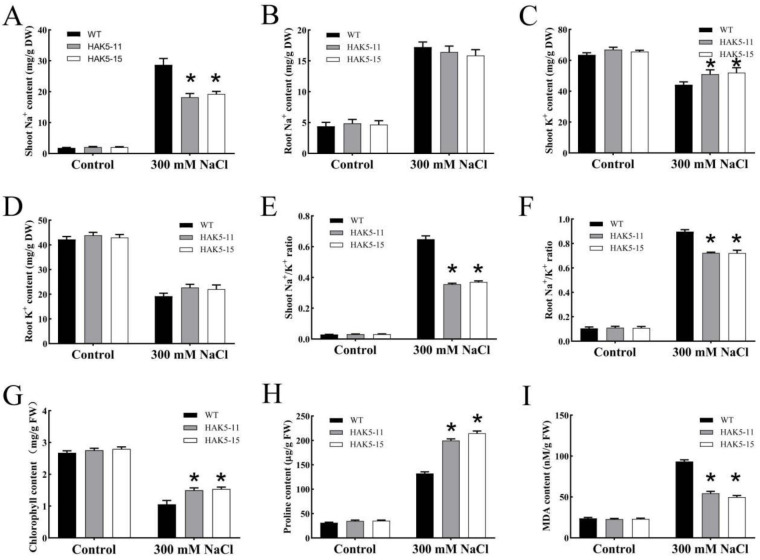
Physiological parameters of WT and transgenic plants grown in the soil under salt stress. Five-day-old seedlings of WT and the two transgenic lines HAK5-11 and HAK5-15 were transferred to soil and allowed to grow for 8 days, then treated with or without 300 mM NaCl. Selected physiological parameters were measured on the 21st day after salt treatment. (**A**,**B**) Determination of shoot and root Na^+^ and (**C**,**D**) shoot and root K^+^ contents in WT and transgenic plants grown in soil under salt stress for 21 days. (**E**,**F**) Na^+^/K^+^ ratios in shoots and roots of WT and transgenic plants grown under salt stress for 21 days. (**G**) Chlorophyll, (**H**) proline, and (**I**) MDA contents in WT and transgenic plants. Values are expressed as means ± SD of three replicates. Asterisks indicate significant differences between WT and transgenic plants as indicated by Tukey’s test (*p* < 0.05).

## Data Availability

Data are contained within the article and Appendix A.

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
