# Peer review of "Positive Regulatory Roles of Manihot esculenta HAK5 under K+ Deficiency or High Salt Stress"

_plants, 2024, doi:10.3390/plants13060849_

Round 1
Reviewer 1 Report
Comments and Suggestions for Authors
Authors have characterised a a gene coding for K-transported from Cassava plant and analysed its expression in transgenic Arabidopsis plants containing the said gene. The presented work is significant in view of enhancing K-uptake under K-deprived/starved conditions. There are a few concerns that should be addressed before the manuscript is published.
What is the novelty presented in the study? It is important since several genes encoding various members of KT/HAK/KUP have been identified in the past. So, how the new transporter/gene identified in the study is different. Is it only that a new plant has been studied.
How much were the changes in expression level of the identified gene under salt stress or under K-deficiency?
In my opinion, authors must include some quantitative data regarding gene expression levels in the abstract. Better incorporate few lines from conclusion in the abstract. In the abstract, please include information about the background, what is done, methods, findings and conclusion.
What were the sources of plant material used in the study?
How much was the age of cassava seedlings at the time of stress treatment?
Provide relevant details of stress treatment.
Several methodological details are missing.
Section 2.7 and 2.8 Methodological details are missing
Authors should include a subsection where they describe all the biochemicals/chemicals used in the study and their source!
Line 31: how climate change caused salinity? An explanation is required.
Line 35: What is the difference between productivity and yield?
Line 137-138: What does ‘healthy and uniform growth’ mean?
Line 140: samples were harvested at different time intervals. When after the treatment or during the treatment?
Line 140: plated ?
Line 146: how much was PPFD?
Line 147-148: size of pots? Weight of medium?
There are repetitions regarding RNA extraction in sections 2.3 and 2.3!
Comments on the Quality of English LanguageModerate revision is required.
Reviewer 2 Report
Comments and Suggestions for Authors
General Comments:
In this manuscript, the authors have studied the HAK5 gene of cassava and analyzed its expression pattern under low potassium and high salt stress conditions. They functionally characterized HAK5 through yeast complementation and overexpression studies. However, authors need to clarify several doubts related to the manuscript before publishing it. First, the HAK5 gene was already identified from the genome assembly of cassava and its expression pattern under different abiotic stress conditions by Ou et al. (2018) (https://doi.org/10.3389/fphys.2018.00017). Why did the authors not use the already-identified HAK5 gene for their experiment? What is the reason the authors redesigned the HAK5 gene for their study? I could not understand why the authors created the phylogenetic tree. For example, more than 20 HAK family genes were identified in many plants, including cassava. But the authors used only the HAK5 gene of cassava for the construction of the phylogenetic tree. Hence, authors should recreate the phylogenetic tree with more cereals (including other cassava HAK proteins). The simple clustal alignment was not used for further research. For example, crystal structure is currently available for HAK family transporters (https://doi.org/10.1038/s41467-020-14441-7). Hence, authors should analyze the key functional residues involved in K+ binding, transport, and more by homology modeling and structure for the HAK5 protein. Authors can refer to the Ceasar et al. 2023 article (https://doi.org/10.1080/07352689.2023.2243108). Because the bioinformatics section is very low in the present manuscript, how did you select low potassium (50 µM) and high salt concentration (150 mM) levels for this experiment? How many-day-old plants were selected for stress experiments? Why the authors did not analyze the K+ and Na levels in transgenic Arabidopsis under salt stress (75 mM and 300 mM) Therefore, authors should assess the uptake of K+ and Na in transgenic Arabidopsis under 75- and 300-mM salt stress conditions. The authors analyzed the HAK5 expression pattern under 150 mM concentration, but they did not analyze the transgenic plant growth under 150 mM concentration. What is the reason the authors selected 75 mM and 300 mM for further analysis? Why did the authors not analyze the expression pattern in transgenic Arabidopsis? What is the reason the authors selected Arabidopsis instead of cassava for the overexpression experiment?
Specific Comments:
1. L. No. 12. Authors should write the full form of HAK/KT/KUP at first use and rectify this error in the entire manuscript
2. L. No. 35. kindly expand SOS1 and all other abbreviation at first use
3. l. No. 90. HvHAK1 denotes gene so should be kept in italics. Check for such errors at other places too.
4. L. Nos. 190-191. Needs more details. Is it a translational fusion with GFP? What was the promoter? Plasmid map??
5. L. Nos. 210-211. What was the promoter for MeHAK5? The authors should include plasmid maps for all constructed plasmids in suppl. files.
6. L. No. 212; authors should change “transformed” into “mobilized”. Because, mobilized is the correct term.
7. L. No. 287. under what growth condition?
8. Figure 2B & C lack statistical analysis, but the authors have claimed as “significantly” expressed in the text? They should include the stat. analysis.
9. L. No. 304. Check the plasmid name under heading 3.3. pCAMBLA1300-MeHAK5: GFP? but different name in methods?
10. L. Nos. 325-326. Results are misleading! What is the Km value? One cannot assay affinities based on simple growth? but need to estimate the amount of K+ transport.
11. L. No. 328. Why and how affinity is calculated in complementation assay in results?
Comments on the Quality of English LanguageMostly fine
Reviewer 3 Report
Comments and Suggestions for Authors
My comments can be found in the attached PDF.

Overall, while the English proficiency is evident, minor adjustments in sentence structure, phrasing, and clarity could further enhance the quality of the writing.
Round 2
Reviewer 1 Report
Comments and Suggestions for Authors
The manuscript has been considerably revised. However, I have a few minor suggestions for the improvement of presentation.
These are:
Latin names should be italicized throughout the manuscript, including citations.
Arabidopsis thaliana is written twice in a single sentence. It should be corrected.
I have seen several recent articles on salinity tolerance in crop plans and expression of related genes by Dr. Suriyan Chaum and his group (from NSTDA, Thailand). I was surprised that not even a single article, which primarily are on the same theme, has been cited.
Comments on the Quality of English Language
A minor editing / corrections is required.
Reviewer 2 Report
Comments and Suggestions for Authors
The reviewers have appropriately addressed all the concerns. It can be considered for publication.
Comments on the Quality of English LanguageEnglish looks fine.
Author Response
Thanks very much for your positive comments.